



# A Close Observation to a Typical Continental Valley Glacier Surge in Northeastern Pamir

Xin YAO[1], Fuchu DAI[2], Iqbl. JAVED[3], Lingjing. LI[1], Zhongsheng WANG[1], Sheng LING[1], Zhengkai ZHOU[1]

1) Key Laboratory of Neotectonic Movement and Geohazards, Ministry of Land and Resource, Institute of Geomechanics, Chinese Academy of Geological Sciences, Beijing, 100086

2) Beijing University of Technology, Beijing, 100124

3) Department of Earth Sciences, Abbottabad University of Science and Technology, KPK, Pakistan, 22010

*Correspondence to*: X. YAO (yaoxinphd@163.com)

**Key Points:**
- ➢ A close expedition and multi-temporal RS interpretation discovered a rare continental glacier surge.
- ➢ It has significant observation difference between quiescent phase and surging phase.
- ➢ Relatively higher ice choking uplift and large chaotic crevasses area extent, but having small integral movement distance with low velocity are its characteristics as compared with oceanic glacier surging.
- ➢ Stagnant downstream tongue and thick superglacial moraine mostly contributed to this type surge's features.

## Abstract:

In May 2015, it is reported that "a glacier slid 20 km long, accompanying with disappearance of ~10 km$^2$ grasslands on the north slope of Mt Jiubie, Kongur Mountain, northeastern Pamir, China". Based on expedition and multi-temporal RS image interpretation, this paper confirms that it was a rare continental valley glacier surge occurred in the tributaries of Kalayayilake Glacier, and find: (1) It has significant phenomena difference between quiescent and surging phase, including distorted medial moraine ridge, extruding-bulging ice masses, disappeared superglacial lakes etc; (2) Compared with the oceanic glacier surge, its characteristics are higher ice choking uplift and large chaotic crevasses area extent, but having relatively small integral movement distance with low velocity; (3) Environmental factors of large glacier coefficient, long tongue, low altitude, especially the stagnant downstream tongue and thick superglacial moraine, contribute to continental glacier surge's features, and the long-term temperature rise and rainfall enhancement in study area seem to be consistent with the occurrence of this surge; (4) This surge brings out severe strength reduction of glacier, rapid ablation of ice, congestion in the subglacial passageway, and accumulation of englacial water, which together have being bred the risk of terminus advance suddenly to result in flooding and debris flow.

**Keywords:** Glacier surging; Continental glacier; Northeastern Pamir; Glacier disasters; Geo-hazard

## 1. Introduction

Glacier surge generally occurs when the ice tongue moves at a velocity several times the normal velocity for days or years (Meier and Post, 1969), accompanying with many extreme phenomena. Surge is the "earthquake" in the glacier regime and it has a great significance for complete and deep understanding of glaciers. The moisture-sufficient oceanic glacier surges have already been reported extensively, such as Black Rapids Glacier and Variegated Glacier in Alaska, North America (Kamb et al., 1985); Trapridge Glacier and others in the St. Elias Mountains, Yukon Territory, Canada (Frappe and Clarke, 2007; Clarke et al., 1986); polar glaciers In Iceland (Jónsson et al., 2014); Midui Glacier and Zelongnong Glacier in eastern Himalayan (Zhang, 1983); dozens of glaciers in Karakorum (Shi and Zhang, 1978; Yao et al., 2004;





Kotlyakov et al., 2008; Copland et al., 2011; Quincey et al., 2011, 2015); transitional glaciers from the
oceanic type to semi-continental type in Shaksgam River (Mustagh river) areas; north slope of Karakorum
Mountains (Shangguan et al., 2005; Liu and Wang, 2009).
46       While in the cold and arid inlands (such as northern margin of Tibet Plateau) and high mountains of
central Asia (Tian Shan), continental glacier surges have been hardly observed. There were only some cases
about relics or suspected phenomena reported, and lacking of detail describes and expeditions (Li et al., 2013;
Niu et al., 2011; Guo et al., 2012; Holzer et al, 2015), which made the less understand for this type surge.
50       In May 2015, herdsmen, residing near Kelayayilake Glacier, Xinjiang province, China (Fig. 1), stated that
the large-area grasslands disappeared and replaced by numerous "fierce-looking and saber-rattling" black ice
masses, which was reported by incomplete realization and extensively informed to the public about "the
glacier slid 20 km and destroyed ~10 km$^2$ grasslands". This paper confirms that it was a very rare continental
glacier surge, and presents a close and dynamic observation based on field investigation and interpretation of
multi-temporal RS images, gains some new finds.
# 2. Glacier environment
## 2.1. Geological setting

58       The Kelayayilake Glacier is located on the north slope of Mt. Jiubie, Kongur Mountain, northeastern
Pamir, with a peak 7649 m above sea level (a.s.l.) and valley-mouth of 2485 m a.s.l.. In , the southeast side,
about 50 km away, is the famous Muztag Ata (7546 m a.s.l.), and in the northwestern is the 5753 m a.s.l.
Kungai Mountains. These high mountains tectonically resulted from the combined action of thrusting uplift
of north Pamir fault as well as the extensional orogenesis of Kongur normal fault at the south side (Robinson
et al., 2004). The Gez active fault, passing through Kelayayilake glacier, is NWW-trending, 190 km long and
dextral with inclination direction of 230° and inclination angle of 40°-76° (Fig. 2 A), and its active triggered
an *Ms* 6.4 earthquake on Nov. 15, 1959. The lithology is mainly Lower Proterozoic (Pt1) hypometamorphic
schist. Between Kungai Mountain and Kongur Mountain, the SW-NE flowing Gez River crosscuts
northeastern margin of Pamir Plateau, and into which Kelayayilake Glacier drainage converges (Fig. 1).
## 2.2. Hydroclimatology condition

69       The study area is an extreme continental glacier zone with very low temperature, and its precipitation
mainly comes from the mid-latitude westerly circulation and local circulation (Shi, 2000). The nearby Gez
River valley has annual average temperature lower than 5 °C, whereas the temperatures in mountainous areas
range from -27.2 to 32.7 °C. The annual precipitation ranges from 60 to 120 mm (Mao et al., 2006; Shi et al.,
2006). Due to the large altitude difference, topographical complexity and climate difference, the
spatio-temporal distribution of precipitation is non-uniform. The precipitations increases with the increase in
elevations, and the annual precipitations near the equilibrium lines (~4200 m a.s.l.) is 477-679 mm (Su, et al.,
1989). At regional scale, the annual temperatures and precipitations in south Xinjiang and northeastern Pamir
have been increasing since 1960s. The watershed is getting significantly wetter and warmer by late 1990s,
and the river runoff volumes in recent 47 years have been rising by 3.0%/10years (Mao et al., 2006). The
average annual precipitations in southern Xinjiang increased by 32% from 1956-1986 to 1987-2000. The
average temperatures in Gez River watershed in recent 40 years increased by 0.23 °C/10years (Xue, et al.,
81    2003).





**Fig. 1 Location, topographical characteristics of glacier, region of two surging tributaries and profiles positions, back ground is 1 m resolution remote sensing image**



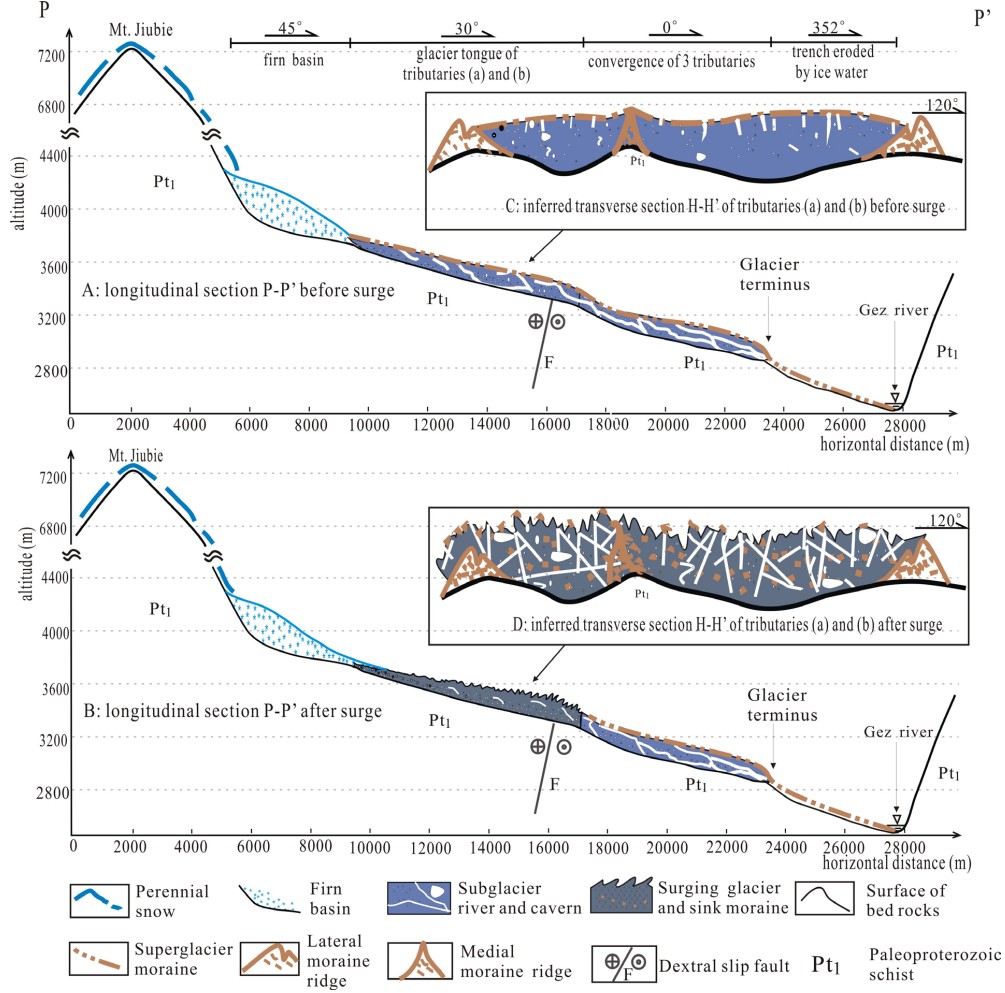

**Fig. 2 Sections of Kelayayilake Glacier before and after surge**

## 2.3. Glacier parameters

Kongur Mountain together with Muztag Ata and Kungai Mountain are the barriers between Pamir Plateau and Tarim Basin which effectively block the moist circulation from the west and guarantee the favorable condition for development and growth of modern glaciers in this region. The modern glaciers are radial and feather-like glaciers distributed around Alps and ridges, mostly at an elevation of 4000 m a.s.l. and above. The general snow lines are at altitude of 4200-5200 m a.s.l.. As typical continental glaciers, their velocity and melting are relatively low.

The Kelayayilake Glacier, 20.3 km long and covering 128.15 km$^2$ with an ablation area of 47.1 km$^2$, is the largest modern glacier on the north slope of Kongur Mountain. It has a glacier coefficient RAA (Ratio of Accumulation area to Ablation area) of 0.63, with an equilibrium line elevation ~4200 m a.s.l., glacier terminus of 2780 m a.s.l. and having upper limit of firn basin is up to 5300 m a.s.l.. The glacier consists of three major tributaries (the west, the middle and the east marked as (a), (b) and (c) respectively), and due to





the slow movement, two neat Medial Moraine Ridges (MMR) are formed (Fig. 1 C, Fig.2 C). The smallest
tributary (a) is 20.5 km long and 300 m average wide, with an ablation area of 6.3 km$^2$; tributary (b) is 19.7
km long and 500 m wide, with an ablation area of 10.7 km$^2$; tributary (c) is the longest one, which is 21.4 km
long and 1500 m wide, with an ablation area of 30.1 km$^2$. Surge occurred at tributaries (a), (b), and the
confluence zone of three tributaries (Fig. 1 C). Longitudinal section of the major surging tributary (b) shows
that the firn basin is ~4 km long and 400 m high, having tongue section ~8 km long with 400 m altitude
difference, following the downstream trunk section ~6 km long and dropped 300 m altitude(Fig. 2 A). The
upstreams of tributary (b) and (a) were convex which seem to be an ice reservoir, and the downstreams were
concave which seem to be an ice receiving area (Fig. 2 A).
The superglacial moraine had well developed from trunk terminus upward to 10 km long, which is an
important characteristic of the continental-type glacier. Its thickness gradually decrease from the downstream,
average thickness of ~0.3 m and maximum of 2-5 m (Fig. 3 P1), to upstream. Glacial cliffs, about 10-300 m
long and height difference of 5-40 m, were well developed in the midstream, at their feet often accompanied
with superglacial lakes with diameter ranging from 10 m to 200 m. The covered debris thickness in the
upstream were less than 0.3 m, and flow-textured crevasses were intensely developed. The height differences
of glacier surfaces varied greatly and were up to 20 m. The micro-morphological shapes were
well-developed, and glacier fissures were interconnected, mainly characterized with englacial and subglacial
ablations as well as subglacial rivers. These parameters indicate that it belongs to Tuomuer-type continental
glacier (Mt. Tomor Glacier Research Team, CAS, 1982).

## 3. Scenes of glacier surge

On May 19, 2015, the glacier surge was investigated along the northwest Lateral Moraine Ridge (LMR)
of tributary (a) (Fig. 1). The trunk section between terminus cliff and tributaries confluence was same as
previous scenes (Fig. 3 P1), however the tributary (a) showed some amazing phenomena: the "black ice"
fragmented expansion interbedded with debris in the glacier trough, which puncture through the overlying
moraine layer to uplift 5-40 m high, and some parts were extruded out of the trough. The broken ice masses
were pressed to override LMRs, forming "aground ice river". Hence, the debris-covered ice masses were
directly irradiated and more largely faces exposed to air, which accelerated the ablation to form ice forests.
The crests of some ice columns lifted the erratic moraine boulder, and the lower crevasses (mostly 1-10 m
wide) were intersected vertically and horizontally. The melt water on ice block surfaces dripped down and
flew around, making "rumbling" sound. Moreover, the ice forests and ice blocks collapsed continuously,
together with moraine deposits and fell into crevasses. When they overflowed the LMR, the
extruding-bulging ice masses collapsed and rolled down along the surfaces of LMR slope, and rapidly
melted off, leaving a small moraine heap. Generally, the extruding-bulging scenes were most severe at the
left mid-stream concave bank of tributary (a), and were gradually weakened towards up- and downstream
(Fig. 2 B, D). To systematically and fully display this fractured extruding-bulging, we selected 5 typical
scenarios from downstream to upstream (Fig. 3 P1 to P5), with corresponding positions and mirroring
directions shown in Fig. 1C.





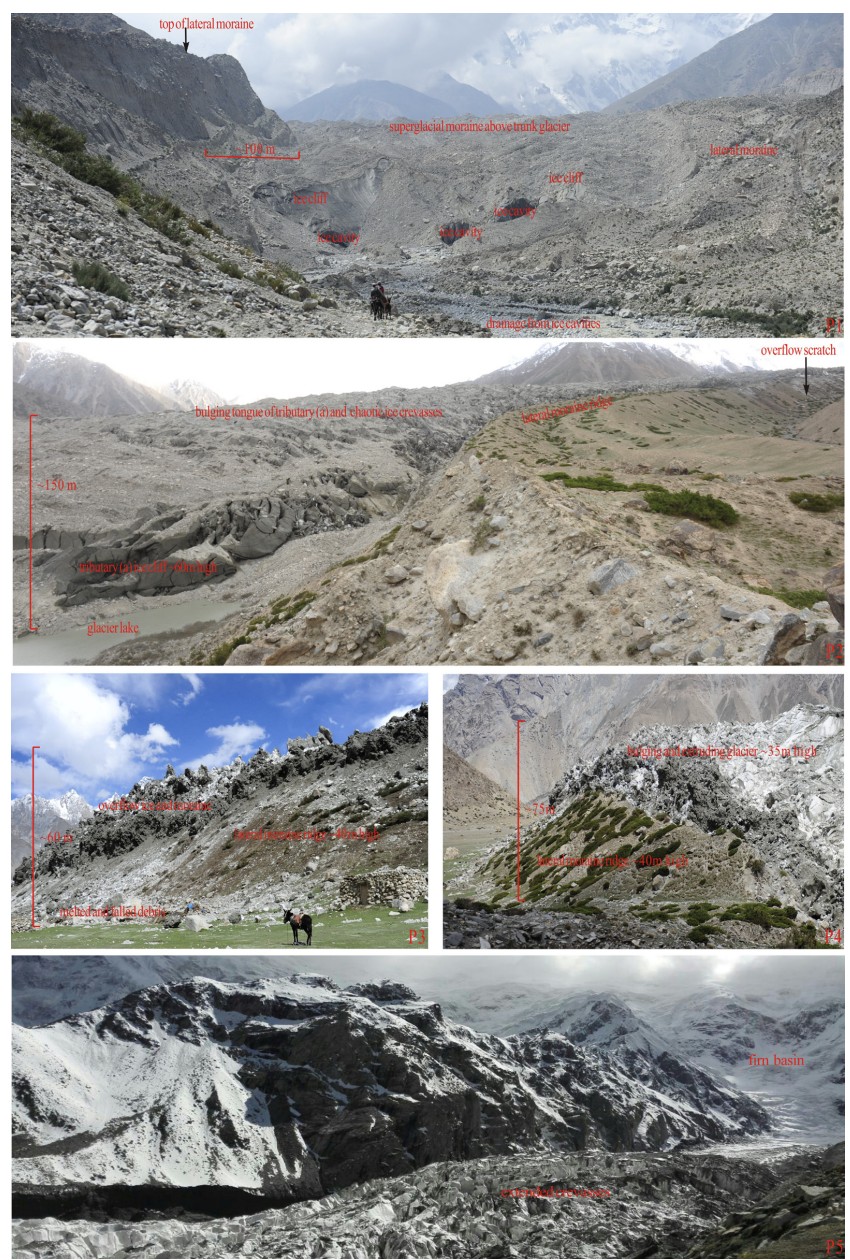

**Fig. 3. Scenes of surge at Kelayayilake Glacier**

**P1** Undisturbed terminus of trunk glacier, 40-m-high and 200-m-wide glacial cliff, in which there were two 40-m-wide ice caves; water burst from the west one at a rate of ~ 1-5 $m^3$/s, whereas no water from the east one. Surface covered 0.3-3 m thick debris, and occasionally found crevasses and ice wells; mirror direction 180°. **P2** Terminus and downstream of tributary (a), ice mass extruded, uplifted, rushed through the overlying debris, formed chaotic crevasses, which directly reflected the causation of the disappearance of grasslands. The left-terminus of glacier (a) melted, forming 50-75 m high glacial cliffs and lakes. On the overall, mid-ridge uplifted above the lateral moraine ridge; mirror direction 145°. **P3** Middle midstream of tributary (a), ice mass uplift and overflowing were most serious, and the extruded lower glacier masses fractured and expanded; black ice masses spilled out ~10 m above the 45-m-high lateral moraine ridge and were partially suspended outside the ice moraine ridge, starting to melt and fell down and endangering the grasslands and houses below it; mirror direction 61°. **P4** Transverse section of upper midstream of tributary (a), glacier body expanded, fractured, wavy lifted, and extruded to overlap the ~20-m-high lateral moraine ridge. mirror direction 45°. **P5** Source of glacier (a), ice mass flew out of the firn basin, mirror direction 250°.





# 4. Formation and development of surge

Here in-situ investigation and multi-temporal RS images interpretation were carried out to determine the initiate time, process, deformation characteristics and velocity of glacier surge etc., which help us to understand Kelayayilake Glacier surge mechanism and evolution.

The RS data includes 8 periods of 15-m resolution ETM images, 3 periods of 2 m GF1 images, and 1 period of 1 m QuickBird (QB) image. The data of glacier ablation is the qualitative observations of subglacial river water flow at trunk glacier terminus by local residents. On the other hand, the phenomena of glacier changes are the field observations. The time series of these data are listed in the axis of Fig. 4.

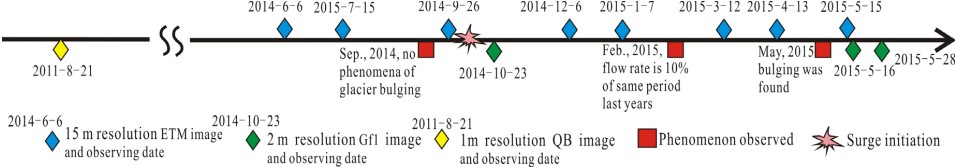

**Fig. 4 Time series of observations for glacier surge**

## 4.1 Development process

Glacier surge not only led color differences in surface debris, dirty ice, snow, superglacial lakes and crevasses, but also generated plasticity extruding & bulging textures. Using multi-temporal observations (Fig. 5, Fig. 6, Fig. 7), we can divide the development process into five stages: inoculation → triggering → extrusion → surge → fracturing.

**(1) Inoculation**: 2011-8-21 QB and 2014-6-6 ETM revealed that the mid- and downstreams of both tributaries (a) and (b) were dominantly covered by moraine and only locally exposed ice showed the continuous flat textures, while tributary (c) was coarse-surfaced and covered throughout with glacial cliffs and superglacial lakes. The 2014-7-15 ETM image shows coarsened fluid textures in tributaries (a) and (b), but no obvious surface changes in tributary (c). These phenomena indicate that Kelayayilake Glacier had slow body movement and more stable surface morphology compared with oceanic glacier, but movement of tributaries (a) and (b) are relatively faster and unstable than that of tributary (c).

**(2) Triggering:** 2014-9-26 ETM and 2014-10-23 GF1 show that the "black ice" areas at midstreams of tributaries (a) and (b) were expanded and the scope of fluid coarse textures extended to the downstream, while the midstream moraine showed horizontally extrude-deformed textures, however the surface textures in tributary (c) were still coarsened, indicating the summer high temperatures led to the ablation of exposed ice masses and promoted the surge of tributaries (a) and (b). However, at this moment no ice bulging was observed at the LMR, because by the end of September, 2014 when the summer grazing ended, the herdsmen did not notice any glacier uplift when they departed from the neighborhood of this glacier.

**(3) Extrusion.** The 2014-12-06 and 2015-01-07 winter ETM images show that the exposed ice and superglacial lakes were frozen, and the low-lying places were covered by snow. The horizontal extruded textures at the midstreams of tributaries (a) and (b) became coarsened, likely the internal flow and extrusion stress strengthened significantly in the ice masses, preliminarily suggesting the surge blocked and extruded to uplift. Though covered by fresh snow, the 2015-03-12 ETM images show obvious extrusion with two evident characteristics: first, the textures of tributaries (a) and (b) were significantly darkened and started to horizontally expand, and the MMR was distorted in several positions; secondly, the front of the surge extruded textures reached the confluence with tributary (c) and got blocked, forming the extrusive uplift





scarps, from which the surge scope could be outlined. At that time (Feb. 2015), the subglacial flow at the trunk terminus suffered a large reduction of about only 10% as that of previous years at the same time, and the flow was muddy, which indicated the alteration of subglacial water circulation.

**(4) Surging.** 2015-04-13 image shows that the extrusion of tributaries (a) and (b) propagated fast with significant lateral expansion, and the trajectory of MMR was further distorted. Tributary (a)'s left LMR was destroyed and became discontinuous. On the surface of three tributaries confluence zone and the downstream, the superglacial lakes displaced at a short distance, and extruding-bulging ice masses were melted at a large areas.

**(5) Fracturing**. 2015-05-15 ETM and 2015-5-16 GF1 showed the exposed ice masses were severely melting, fracturing and extruding. The mid- and downstreams of tributaries (a) and (b) were covered with "black ice", the arc-shaped horizontal compressed and short longitudinal tensional fissures were intersected vertically and horizontally, wavy distributed and spread on glacier surfaces. The ice masses were swollen, uplifted and "spilled out" of LMR. The melted ice clumps and moraines collapsed and rolled out of the LMR, forming the shoveled, melted and corroded scratch. The low-lying inner LMR was scraped and washed out the debris to the outer LMR (Fig. 2 D, Fig. 6 C-D), these scenes corresponds to Fig. 3 P2-4 as well. The MMR between tributaries (a) and (b) was further contorted, and some sections even disappeared. Notably, the tributary (c) at the confluence showed regular distribution and short extruded longitudinal tension crevasses, indicating this region had been the main anti-slip section and started to endure the major extrusion stress from the surge of tributaries (a) and (b).

Based on all evidences, we inferred that the low velocity glacier had been experiencing long-term stress and strain accumulation, and the surge initiated in Oct., 2014, grown in winter of 2014-2015, and fully broke out in the spring and summer of 2015.

## 4.2 Surge velocity

Unlike oceanic glaciers' evident rapid propulsion at the terminus, Kelayayilake Glacier surge was mainly characterized by extruding-bulging and uplift, hence the overall displacement was difficult to measure. We compared the position changes of featured points in mid- and downstreams of tributaries (a) and (b) to calculate the surge velocity, including the terminus (point $a$ in Fig. 6 A-D, Fig. 7), LMR (Fig. 6 A-D, pint $b$, $c$), extrusion fracture (Fig. 6 A-D, point $d$) and MMR (point $e$, $f$ in Fig. 6 E, F).

(1) The displacement from Aug. 21, 2011 to May 16, 2015 was 100 m, judged from relative positions between the terminus glacial cliff of tributary (a) (Fig. 6 B) and a meadow on the LMR. The MMR between the terminus of tributary (b) and tributary (c) moved ~250 m downward (Fig. 6 C, D), by the condition of gentle slope and blocking by tributary (c). From this, we deduced that at the mid- and upstreams was larger than 200 m.

(2) From May 15 to 28, 2015, the surge front of tributaries (a) and (b) elongated 30 m (2.3 m/day), about 8-fold as that of regular velocity, indicating the accelerated trend of development.

(3) Compared with 2014-10-23 (Fig. 6 E) and 2015-5-28 (Fig. 6 F) images, the MMR between tributaries (a) and (b) (point e in Fig. 6) shifted sideway up to ~200 m, and the point $f$ almost disappeared. Besides, ETM images on 2014-9-26 (Fig. 5 A) and 2015-5-15 (Fig. 5 F) show that the MMR at point $e$ moved laterally to a similar distance.

Thus, the advance distance estimated at the terminus of tributaries (a) and (b) was ~200 m. From October 2014 to May 2015, the average velocity was about 1.0 m/day. The displacement and the velocity of Kelayayilake Glacier is less as compared to those of oceanic glaciers (distance 1-11 km, velocity 0.15-6 km/day); however, the propagation of fractured extruding-bulging was up to 33 m/day up- and downstreams



which resulted in complete fracturing of 7-km-length tongues within short time.

## 4.3 Volume

For estimating glacier thickness and volume is uncertain, different types of methodologies were developed,
the common used are Area-Volume, Slope-dependent, and Ice-thickness distribution (Frey et al., 2014;
Haeberli et al., 2016). Subject to the available data, the Area-Volume were tended to use, for which there are
also three sets of scaling parameters (Chen and Ohmura, 1990; Bahr et al., 1997; Su et al., 1984), which
response to three volumes of 0.97, 1.15, and 1.32 billion m$^3$ respectively for this glacier. Though the Su at
al.(1984)'s is the largest, we are prone to use this one for three reasons: (1) The bigger volume seems to be
more reliable, as the mean ice thicknesses are significantly higher in the Karakoram (94–158 m) than in the
Himalayas (54–83 m) (Frey et al., 2014), and our study area is adjacent to Karakoram. (2) This set of
parameters is deduced from the glaciers in Tian Shan which is similar to our study glacier, as well as all of
them are continental type glaciers. (3) The parameters are the only available area-related parameters that
exist for high Asian glaciers.
The surge area of tributaries (a) and (b) was about 13 km$^2$. According to the statistical relationships
between glacier area with its thickness and volume (Su et al., 1984), estimated:
$$H_g = -11.23 + 53.21 A_g^{0.3} \quad \text{(1-1)}$$
$$V_g = 0.0674 A_g^{1.16} \quad \text{(1-2)}$$
$A_g$ is the glacier area, the thickness $H_g$ = 103.6 m, and the volume $V_g$ = 1.32 billion m$^3$.
In-situ expedition for tributaries (a) verify that the thicknesses are ~32 m and ~ 75 m at the banks of
upstream and middle-stream, and 110 m at the terminus; the height of cliffs at trunk terminus was ~ 40 m.
Commonly in case of U-shaped trough bottom, the thicknesses at the middle parts of cross sections should
be larger and may be referred to the average thickness (~110 m) of nearby similar continental glaciers in
Tian Shan (Yang and An, 1991), the thicknesses and volumes of tributaries (a) and (b) estimated from
equations 1-1 and 1-2 are acceptable.

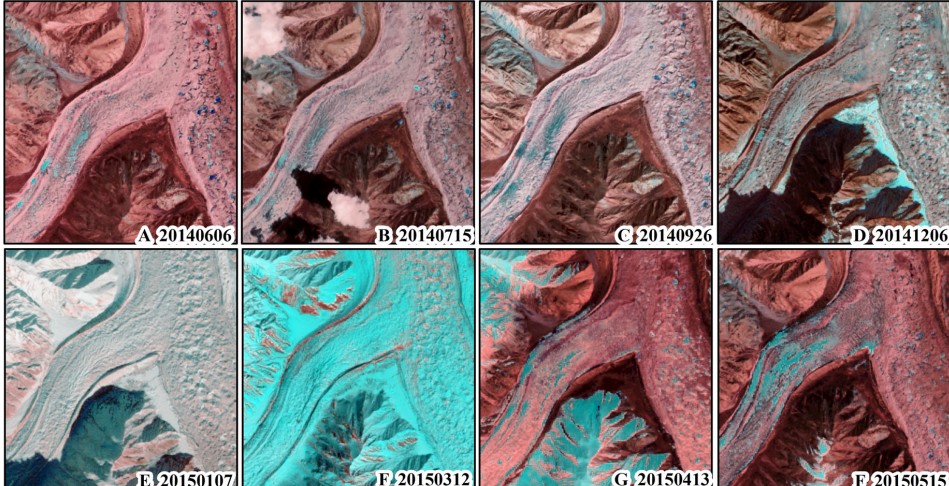

**Fig. 5 Eight phase ETM RS images covering zone of mid- and downstreams of tributaries (a) and (b)**





**Fig. 6 Characteristics of glacier surge from high-resolution RS images**



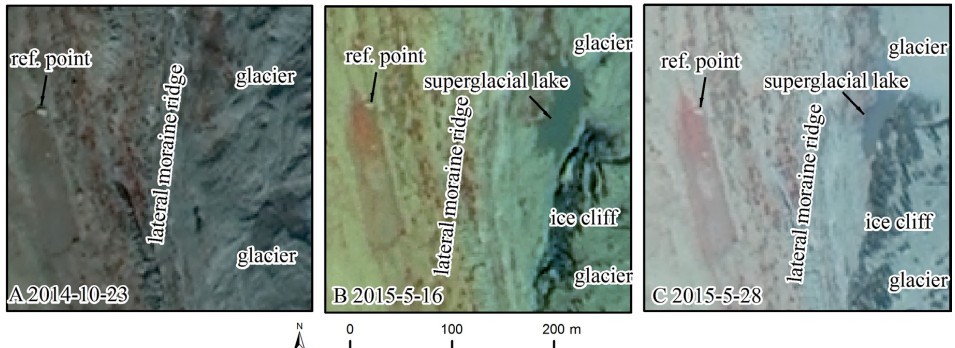

259                      **Fig. 7 Movements of glacial cliff and ice lakes at point *a* in Fig. 6**

# 5 Analysis and discussion
## 5.1 Environmental factors
Like any geo-phenomena, the occurrence of glacier surge depends on some environmental factors
(relatively static state) as well as the triggering factors (relatively dynamic state).
The major environmental factors are stability coefficient (Yao, 1987) and geometrical shape (Clarke,
1991). Multivariate analysis indicates that surge-type glaciers are characteristically long, wide, and
low-altitude tongue (Barrand and Murray, 1991). The critical glacier coefficient (0.63) , ~13 km$^2$ tongue area
and relatively lower tongue elevation (2780 m a.s.l. at terminus) of Kelayayilake Glacier are capable of
inducing a surge. And tributary glaciers have a greater tendency to surge than trunk glaciers, presumably
because they may themselves be surge-type and may additionally participate in surges of the trunk glacier
(Clarke et al., 1986).
The velocities at firn basin and upstream were up to 100-1000 m/year, however the velocity at the trunk
terminus was less than 10 m/year in previous years (Jiang et al., 2014). Together with the "steep upstream
and gentle downstream" terrains, the moving glacier was blocked and extruded, forming a situation of "push
from behind, block from front". As a result, the midstream and downstream ice masses fractured and uplifted
which provided favorable conditions for the expansive ice masses to spill out of the trough and ultimately
propagate to the upstream and the downstream. The downstream of tributary (a) is located on the concave
bank side, where it became a main section of ice mass spillover.
In addition to this, thick moraine cover was another important factor for the special surging phenomena.
Surface debris did not fully deposit from the surface ice ablation, and about 1/3-1/2 extruded from the
englacial when the internal and bottom debris moved along the ice layers or fractured surfaces (Mt. Tomor
Glacier Research Team, 1982; Shi et al., 1989), suggesting that the Kelayayilake Glacier has an active
internal movement which contributes to stress accumulation.
Surface debris has a double effect on glaciers ablation. (1) Debris layer can significantly reduce surface
reflectance and promote the absorption of radiant heat. When debris layer is thicker than 3-8 cm, it has a
very small heat capacity and the moisture inside the thin-layer debris layer has high thermal conductivity,
resulting in promotion of the ice surface ablation (Wang, 2011; Ding et al., 2014; Liu, et al., 2013); (2) on the
other hand, when its thickness is very large, the downward conducted heat quantity was largely consumed by
the surface gravels before entering the deep gravels or underlying ice surfaces, hence the surface debris
inhibited the ice ablation and thus enhanced the glacier activity (Kang et al., 1985; Bookhagen and Strecker,
2011; Xie, et al., 2004; Han, et al., 2005).
As mentioned in section 2.3 of superglacial moraine characteristic, the ablation strength of tributaries (a)
and (b) was highest in the lower midstream, and weakened upwards with the altitude rise and temperature
reduction, and weakened downwards with the increase of debris thickness. The ~0.3 m thick debris layer



helped to preserve the glacier activity which was a key environmental condition that produced severe
extruding-bulging at the mid- and downstreams, and was also the basis which make it different from oceanic
glaciers.

## 5.2 Triggering factors

Two major types of triggering mechanisms for glacier surge initiated had been concluded (Robin, 1969).
One is due to the temperature instability to promote basal glacier deformation and porosity, with a positive
feedback among pore water pressure, deformation, and basal flow(Clarke et al., 1984; Murray et al., 2000),
which is characterized by a relatively slow velocity and an initiation phase that lasts several years before the
peak of the surge is reached and a termination phase that comprises several years of deceleration following
the peak of the surge. These surges have been observed to begin their acceleration/deceleration independent
on any seasonal control. (Murray et al., 2003). Another is due to subglacial pore water pressure instability,
triggering the flow instability (Kamb et al., 1985; Björnsson, 1998). Such water pressure surges are
characterized by rapid acceleration and deceleration (i.e., days to weeks long) (Kamb et al., 1985) and likely
to initiate during winter season, when the drainage efficiency is low, and terminate during summer season,
when the drainage efficiency is high (Burgess et al., 2012; Fatland and Lingle, 2003). Commonly, both type
of surge were observed at oceanic glaciers, for the other type of surge, such as those at Karakoram region,
they are heterogeneity of triggering mechanism (Quincey et al., 2015).
Though Gez active fault passes through glacier and induced big earthquake in the past, however no
earthquakes of $\geq M$s 4.0 ever occurred within the 100 km radius of the glacier, since June 1, 2014, and the
maximum magnitude within 10 km from the glacier was $M$L 1.3 (Fig. 1). Compared with the regional
historical seismicity level, this was a relatively weak seismic period, so the stress instability from the
external could be excluded from our analysis.
From weather station about 10 km away from the glacier surge zone, 2770 m a. s. l., at Gez river valley
(Fig. 1), the daily average temperature was 1.56 ºC, about 0.65 ºC warmer from previous year; the daily
maximum and minimum temperatures were 5.91 and -1.9 ºC respectively, which are about 0.97 and 0.42 ºC
warmer than those of previous year. The accumulative precipitation during the surge occurring and
development periods (from Sep. 1, 2014 to Apr. 28, 2015) was 33.9 mm, which was about 2.3 times that of
the same period last year. The precipitation from 20 rain stations within 200 km of study area during the
period from last winter to Apr. 2015, was about 3 to 4 times larger than the normal levels (private
communication from Xu Baiqing). Moreover, as glacier zones had an apparent vertical climate, the
precipitation above the snow line in the glacier accumulative area increased more significantly (Shi and
Zhang, 1978). The glaciers in Muztag Ata, adjacent to study area (50 km away), have a slight mass gain
contrary to the global trend, but with locally spatial and temporal glacier variations during the last four
decades (Holzer et al., 2015). And for the farther distance, the glacier surges had expanded and increased in
central Pamir since 1990s (Copland et al., 2011). All of these phenomena contribute to the long-period
increase of precipitation and temperature, while a short, local fluctuation in NE-Pamir. Clearly, both
short-term and long-term precipitation and accumulation patterns were favorable for the occurrence
controlled by thermal conditions. On the contrary, the characteristics of surging initiate season, acceleration
and deceleration, as well as velocity deviated from the conclusions of controlled by hydrological conditions.

## 5.3 Periodicity

Glacier surge is periodical which is about 15 to 100 year, and generally 20-30 year for some oceanic
glaciers (Meier and Post, 1969). However, no reports on the repetition of thick debris-covered continental
glaciers have been found. No data from previous surges are available in study area. However, based on
interviews from the old people in the local village, a sudden flood and debris flow was witnessed at the





confluence site of Kelayayilake Glacier and Gaz River about 100 years ago. "At early June that year, the
downstream at the glacier valley was interrupted, and at late June, a sudden huge flood & debris flow (wave
about 100 m high) occurred, carrying cattle-, goat- or yurt-like stones into Gez River. Numerous trees along
the way were destroyed. The debris flow did not stop until about 70 km away." The floods & debris flows on
the northeastern Pamir were usually caused by the burst of ice lakes. However, according to hydrological and
topographic condition in Kelayayilake Glacier valley (Fig. 1), this glacier region did not like to harbor
forming conditions for ice lakes, and it is more likely to be a sudden burst of surging fractured ice masses. If
this were true, the surge periodicity of this glacier should be a centenary scale.

## 5.4 Disaster risk

Different from most reported surges discharge to the fjords or desolation valleys in the remote polar region
and depopulate zone, Kelayayilake Glacier is located aside the Gez River valley alone where the
China-Pakistan international road passes. There are lots of passengers and property, hence the disaster risk
become the most focus question when the surge occurred.
The glacier surges usually underwent three phenomena: evident MMR folds, ice surface fracturing, and
sudden advance at glacier terminus (Meier and Post, 1969), and they ended by three results: internal advance,
no disturber to the terminus; tongue advance, would made disaster to downstream; the tributary surge dam
will play a import role for the trunk surge (Kotlyakov et al, 2008). Until June 2015, except terminus tongue
advance, all of other phenomena had occurred.
Before June 2015, the drainage from the subglacial river at trunk terminus had significantly reduced and
became muddy compared with the same period of previous years; however the simultaneous ablation raised
dramatically because of the abundant crevasses and missing moraine protection. Based on regular glacier
aquifer yield, it was estimated that the englacial water storage has being increased. The surge also made the
hydro passageway to disturbance, and led to ice cavities congestion. In NE-Pamir, summer usually start from
early April and ends by middle October, so the glacier ablation would continue in the upcoming few months.
All of these may be the harbors risks of flood burst as that of hundred years ago, which thereby leads to the
development of secondary disasters, such as debris flow.

## 6 Conclusions

A close observation by expedition and multi-temporal RS image interpretation show that the
continental-type Kalayayilake Glacier in the northeastern Pamir has a significant phenomena difference
between quiescent and surging phase, including distorted medial moraine ridge, extruding-bulging ice
masses, disappeared supergalcial lakes etc. Compared with oceanic glacier surge, its characteristics are
relatively higher ice choking uplift, large chaotic crevasses interval area extent and centennial-scale
repetition, but having small integral movement distance with low velocity. Environmental factors of large
glacier coefficient, long tongue, low altitude, especially the stagnant downstream tongue and thick
superglacial moraine, contribute to its features. And the long-term temperature rise and rainfall enhancement
in NE-Pamir seem to be consistent with the occurrence of this surge. This surge brings out severe strength
reduction of glacier, rapid ablation of ice, congestion in the subglacial passageway, and accumulation of
englacial water, which have being bred the risk of terminus advance suddenly to result in flooding and debris
flow.





# Acknowledgements

This research is supported by the China Geological Survey (12120114001401) and National Natural Science Foundation of China (40902059). Special thanks to Prof. Hou Chuntang and Prof. Zhao Zhizhong for their valuable suggestions for completing this research works. We will give thanks to Prof. Xu Baiqing for providing the information of temperature and rainfall. And we are thankful to handling editor Tobias Bolch, editor Anna Feist-Polner and anonymous reviewers for their constructive comments which greatly helped to improve the original manuscript and giving the opportunity to extended the revision time.

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
