# Peer review of "A Close Observation to a Typical Continental Valley Glacier Surge in Northeastern Pamir"

_The Cryosphere, 2015_

## Referee Comment (RC1) · D. Shangguan (Referee) · 25 Mar 2016

General comments: This paper examined a typical continental valley glacier surge in Northeastern Pamir, using multi-temporal RS images and their own field surveys. The triggering factors are interesting and important for broad audiences as the authors understand. The manuscripts were overall readable. However, The presented techniques in this paper are not well-established. there are several issues that should be addressed before the manuscript is ready for publication. And the glaciological significance is rather weak.

The first critical problem is the velocity. It is widely known that velocity is important one of factors to identify the glacier surge. Although authors are knowledgeable enough to measure the velocity, they selected those surface features no enough to measure the

velocity. Furthermore, the authors did not even show the velocity during active phase. The present writing is unclear to show the velocity field. The second is volume. Volume change or ice transportation is key signal to show the "reservoir" and "receiving " area. However, the authors just calculated the ice volume. No any information showed the ice movement from reservoir area to receiving area. I have to say the volume section is useless or few useful to this manuscript. The third is term. For example AAR(no RAA); debris covered ice (no Back ice); seracs (no ice forests) and so on. The forth: Development process is the biggest inexplicability.

Specific comments Line 19 slid->move Line 20 Mt Jiubie, Kongur Mountain? Jiubie Peak, Kongur Mountian. The same as line 58 Line 22 the find 1 is common knowledge. It could not be shown as one of results. Please point out which difference it is. Line 24 Oceanic glacier->maritime glacier Line 30 "severe strength reduction of glacier" The result is not shown in your manuscript. Actually, the glacier extent expanded slightly. Line 41 In->in Line 44 Mustagh River-> Yarkant River Line 46 Authors mentioned several times this surge is difference with maritime glaciers. Authors only gave the information about glacier surge in high mountains observed hardly. But what is he difference between high mountains and maritime glaciers?

Line 59 delete "," between "In" and "the" Line 72 "The annual precipitation ranges from 60 to 120 mm" -please provide the observation station information. Line 74 precipitations->precipitation (the same as others); increases->increase Line 95 "RAA"->AAR Line 98 Three major tributaries, why? Maybe four major tributaries is better. Line 99 "slow movement" How much is it? Did you have reference?

Line 118. "Scenes of " ->"Event of " Lines 121-123 it is difficult to be understood. Line 126 ice forests->seracs Lines 169-171 The velocity of tributaries a and b is faster than tributaries C. Do you have any evidence? The surface change of tributaries a and b is little. However, the surface of tributary c is coarse and develops lakes. How do you identify the velocity difference among tributaries a,b,c. Line 184 "were significantly darkend"? what 's your mean. It may be related to snow melt out Line 212-214 Pint-

>point Why do not use cosi-corr to extract the velocity field? Point a, b are in lateral deposition; There is obvious landslide in Point c Point e and f are ok. However, can you identify the active phase of glacier surge?s Line 278 clear "this" Lines 298-310 there are two types of triggering mechanisms. But Authors should point out which one it is in this work? Lines 366-367 This sentence is no significance because it is when the glacier was in quiescent and surging phasing. Authors have to point out what difference it is.

Figure 1 parts of text can not be readable, they should be magnified. Figure 3 text can not be readable, they should be magnified.

[Figure]

---

## Author Comment (AC1) · 24 May 2016

Dear Doc. Shuangguan:

Thank you very much for your detailed and veteran discussion to improve this manuscript (tc-2015-235). We tried our best to improve the manuscript in the light of your suggestions, our reply is given belowïijŽ

(1) As referees repeated emphasis at many sentences, the velocities of surge and quiescent are the worst questions. Answer: The advices of using "cosi-corr" method to measure velocity is a perfect idea, we accepted this advice and promoted it by using the SAR offset-tracking method to extract the velocity, which calculated the more accurate and reliable velocity, and avoid the influence of the clouds comparing with the optical images. The new Sentineal-1 SAR data was used to do a high frequent offset-tracking

measure for ten periods during Feb. 2015 to April 2016, the results can not only get the velocity of surge(up to 10m/d) and infer the surge process. Meanwhile, it is the strong evidence to decide the surge start time.

(2) The second is volume. Volume change or ice transportation is key signal to show the "reservoir" and "receiving" area. However, the authors just calculated the ice volume. No any information showed the ice movement from reservoir area to receiving area. I have to say the volume section is useless or few useful to this manuscript. Answer: The volume is difficult to precise measure. Some researcher to measure it by employing the difference between stereo images pre- or after surge, DEM, topographic map, and field survey et al.. For making up the volume defect in this paper, addition to the previous work, we try to collected another field survey data on Aug. 24, 2015 and interpret a 2-m resolution GF1 optical RS image on Oct. 16, 2015. Those new works as well as the process of surge measured by Offset-tracking SAR method, can be effective to calculate the volume change.

(3). The third is term. For example AAR(no RAA); debris covered ice (no Back ice); seracs (no ice forests) and so on. Answer: Thanks very much for giving the glacier term revise, the most term will be accepted by the authors except the special words will be used according to the context.

ïijĹ4ïijĽThe forth: Development process is the biggest inexplicability. AnswerïijŽAs above stated new study date obtained during the open discussion period, development process will be presented newly.

Some specific comments also will be considered in the revised manuscript. And the authors is grateful to the referee D.Shangguan, and well come other community members continue to give more constructive comments to this paper for well presenting the scarce continental glacier surge.

Please also note the supplement to this comment:
http://www.the-cryosphere-discuss.net/tc-2015-235/tc-2015-235-AC1-supplement.pdf

**Supplement:**

[Figure]

**Fig**.4 Glacier velocities calculated by offset-tracking method using Sentinel-1 SAR data in surge phase from 2015-2-18 to 2016-04-11, and in quiescent phase from 2010-03-04 to 2010-07-20 using PALSAR SAR data, alone the velocity profile in Fig.1. The X-axis distance is corresponding to hatch in Fig. 1. The dots mark the velocity in surge phase, the lines show the velocity in quiescent phase.

---

## Referee Comment (RC2) · Anonymous Referee #2 · 14 Jun 2016

A brief summary This is an interesting submission that presents a surge-type glacier in Northeastern Pamir based on field surveys and satellite images. This paper presents developments of its surface features by a glacier surging. The results are presented clearly and are worthwhile publishing. However, I have reservations about the interpretations of surging process. My broad comments are as follows:

1. The development process of glacier surging should be based on surface features and the surface velocity. Authors showed temporal changes in flow speed along the profile using an offset-tracking method with satellite SAR datasets in the supplemental figure. Please add temporal velocity maps and evaluate its uncertainty. I would recommend that authors revise the development of glacier surging based on changes in the surface velocity and surface features. Also please summarized characteristic of this surging (e.g. the timing of initiation/termination, the duration of the active phase, peak

[Figure]

speed). Which method does you used to convert the offsets to the surface velocity? (e.g. Mattar et al., IEEE TGRS, 1998) The SAR offset-tracking method generates displacements along the line-of-site and the platform direction. An assumption is needs to convert these offsets to surface velocity from one path pair (a ascending/descending pair).

Mattar, K.E. et al., 1998. Validation of alpine glacier velocity measurements using ERS Tandem-Mission SAR data. IEEE Transactions on Geoscience and Remote Sensing, 36(3), pp.974–984.

2. The volume section seems to be few useful to this manuscript. I agree with another referee comment. I would like to recommend that authors might remove the volume section.

3. I would recommend that the authors modify figures. For examples, you should change font/line/marker color and size in Figure 1 and 3, which are difficult to read. In figure 2, how to determine the bed/surface profile and the transverse profile before/after surging? Is there any field measurement or reference for these profiles?

4. Please use a general terminology for surge-type glaciers (e.g. Grant et al. 2009; Jiskoot, 2011)

Grant, K.L., Stokes, C.R. & Evans, I.S., 2009. Identification and characteristics of surge-type glaciers on Novaya Zemlya, Russian Arctic. Journal of Glaciology, 55(194), pp.960–972. Jiskoot, H., 2011. Encyclopedia of Snow, Ice and Glaciers: Glacier Surging (p415-428), Springer.

Less major comments are as follows:

L60-61: "In the southeast. . . Kungai Mountains." These are shown in Figure 1. I recommend removing these sentences to spare space.

L63-64: Please add "the Gez active fault" on Figure 1.

L64: ", and its active" -> ". This fault"

L98: "(a), (b) and (c)". Add "Figure 1". As mentioned above, you should changes font size and color in Figure 1. These are difficult to read.

L119-135: Please modify this paragraph. It seems to be difficult to understand what the authors explain.

L187-189: "the subglacial flow…". How to measure this reduction? Englacial hydrology is an important factor of glacier surging. If possible, please add details.

L206: "the low velocity glacier". Please indicate the speed of the glacier flow.

L228-229: "distance 1-11 km, velocity 0.15-6 km/day". Please add references.

L262: What is a stability coefficient? How to calculate it?

L266: What is "the critical glacier coefficient"?

L266-268: Why these factors are capable of inducing a surging? Multivariate analyses only indicate that a surge-type glacier tend to have a wider width and a longer length.

L316: "From weather station" -> "From a weather station"

L328-332: Which mechanism is responsible for surge triggering of the examined glacier?

L342: "the burst of ice lakes". Do you mean "GLOF (Glacier lake outburst flood)"?

L366-368: These are general characters of a surge-type glacier.

---

## Author Comment (AC2) · 15 Jun 2016

Dear Anonymous Referee #2

Thanks very much for your detailed and sincerely comments on this paper, especially given some specific reference information, to improve this manuscript (tc-2015-235). During the discussion period, authors have done new studies and used new data to promote the paper. A new in-situ investigations on Aug. 2015, additional 5 ETM RS images interpretation updated to 2016-06-02, 1 high-resolution RS images interpretation on 2015-12-18, 9 consecutive phases Synthetic Aperture Radar (SAR) offset-tracking survey from 2015-02-28 to 2016-04-11, were carried out to determine the initiate time, process, deformation characteristics, velocity, duration of this surge. Thus, some detailed finds and deep understands have been gained: (1) This surge experienced about

300 days, initiated in Feb., 2015, fully fractured in the spring and early summer, declined after Aug., and recovered to normal status after Oct., 2015; (2) The peak surge velocity wavy transfers from upstream to downstream with 8-10 m/day during initiation period, and propagation of surface cracks was up to 58.3 m/day, resulted in fracturing of 3.5-km-length tongues of tributary(a) and (b), then peak velocity descended from ~10 m/day to ïijĺ1 m/day in remainder time, which almost made the entire trunk disturbed with anti-press longitudinal crevasses; (3) The height of "receiving" area increased 20~40 m with 2.7-3.6×108 m3 ice transferred from "reservoir", and accumulation time of this volume maybe need half to one centennial in quiescent period; (4) Environmental factors of large glacier coefficient, long tongue, low altitude, especially the stagnant downstream tongue and thick superglacial moraine, contribute to its features; (5) Nevertheless, long-period increase of precipitation and temperature, while a short, local fluctuation in NE-Pamir were favorable for the occurrence, the characteristics of surging indicates that this surge initiation controlled by hydrological instability directly; (6) This surge made internal advance and trunk disturbance, but no effect down to the terminus, therefore, there would be no disasters to downstream. Above conclusions maybe have answered some your suggestions, other are not covered, we will try to complete in the next few days. As you mentioned the transfer volume calculation is difficult during surge, but we attempt give a qualitative result according to the field survey, in order to provide more information for reader, despite it is not very accurate. For the figures, we have modified the labels and size for easily read. We will standardize terminology according to some references. Thanks for patient revision to some little errors, amendment line by line will be made by authors.